# Organic Diode Laser Dynamics: Rate-Equation Model, Reabsorption, Validation and Threshold Predictions

**Daan Lenstra** [1,*] [ID], **Alexis P.A. Fischer** [2,3] [ID], **Amani Ouirimi** [2,3], **Alex Chamberlain Chime** [2,3,4] [ID], **Nixson Loganathan** [2,3] and **Mahmoud Chakaroun** [2,3]

1. Institute of Photonics Integration, Eindhoven University of Technology, P.O. Box 513, 5600MB Eindhoven, The Netherlands
2. Laboratoire de Physique des Lasers, Universite Sorbonne Paris Nord, UMR CNRS 7538, 99 Avenue JB Clement, 93430 Villetaneuse-F, France; fischer@univ-paris13.fr (A.P.A.F.); amani.ouirimi@univ-paris13.fr (A.O.); alexchamberlain.chime@univ-paris13.fr (A.C.C.); nixson.loganathan@univ-paris13.fr (N.L.); chakaroun@univ-paris13.fr (M.C.)
3. Centrale de Proximite en Nanotechnologies de Paris Nord, Universite Sorbonne Paris Nord, 99 Avenue JB Clement, 93430 Villetaneuse-F, France
4. IUT-FV de Bandjoun, Université de Dschang, BP 134 Bandjoun, Cameroon
* Correspondence: dlenstra@tue.nl; Tel.: +31-488-75241

**Abstract:** We present and analyze a simple model based on six rate equations for an electrically pumped organic diode laser. The model applies to organic host-guest systems and includes Stoke-shifted reabsorption in a self-consistent manner. With the validated model for the Alq3:DCM host-guest system, we predict the threshold for short-pulse laser operation. We predict laser operation characterized by damped relaxation oscillations in the GHz regime and several orders of magnitude linewidth narrowing. Prospect for CW steady-state laser operation is discussed.

**Keywords:** optoelectronics; OLED; laser; organic laser diode; laser dynamics





## 1. Introduction

An Organic Diode Laser (ODL) is the lasing manifestation of an Organic Light Emitting Diode (OLED). It represents a promising class of new lasers with foreseen applications in spectroscopy, sensing, environmental monitoring, optical communication, short haul data transfer [1,2]. Since Heeger's demonstration of "plastic" conductivity in 1977 [3], organic semiconductor technology has made a huge step forward. With relatively simple, economic, and environmentally friendly production processes, and virtually unlimited availability of amorphous organic semiconductors, organic optoelectronics has become a large research field for various device types such as organic photovoltaic cells (OPV), organic transistors (OFETs), and OLEDs [4–7]. Developments in OLEDs have resulted in successful applications including lighting or display technologies such as screens for TV and mobile phones, but they have so far been underused in optical transmission systems in comparison to their inorganic counterparts, namely the conventional light emitting diode devices (LEDs).

The ODL will open a new era in the field of lasing. Firstly, because solid-state organic materials, contrary to their III-V counterparts, cover continuously the whole visible spectrum as well as part of the IR and UV spectrum. Secondly, they can be deposited more easily on almost any substrate with less energy consumption for the manufacturing process than conventional epitaxially-grown III-V materials [8]. Thirdly, this new device combines properties from dye-lasers and III-V diode lasers and as such will open new perspectives and potential applications. Fourthly, organic electronics is a low-carbon industry, unlike the III-V industry.

Regarding perspectives, organic materials exhibit dependence of the refractive index on the carrier density different from conventional III-V semiconductors, which is largely

due to the specific mobility of disordered organic semiconductors [9,10]. Therefore, new and interesting dynamical behavior will occur, especially when the laser is submitted to different types of external perturbations, such as optical injection and feedback [11]. Potential applications and new possibilities will be facilitated by the ease of deposition of organic heterostructures on a large variety of substrates including silicon, silica, glass, as well as flexible substrates, and by the availability of an almost unlimited library of electroluminescent organic materials [12].

Until now, solid-state organic lasers have been realized by optical pumping of OLEDs provided with an integrated optical cavity [13–15]. Lasing based on electrical injection appears to be much more difficult, because of gain quenching due to triplet accumulation and absorption from the metallic anode [13]. We will summarize our rate-equation theory for an electrically injected ODL [16], extend the theory to include a detailed treatment of the self-consistent reabsorption, and present simulation results for operation below and above laser threshold.

## 2. Characterization of a Laser OLED

Organic Diode Lasers (ODLs) are organic hetero structures with an integrated optical cavity to enhance the interaction time of photons with the active molecules before they are emitted, thus enabling a sufficiently high level of stimulated emission to generate laser light. A schematic of the layer structure of an organic hetero structure is given in Figure 1a. An important difference with conventional III-V semiconductor devices is that the charges are not just electrons and holes, but rather electron-like (i.e., negatively charged) polarons and hole-like (positively charged) polarons. These polarons are special organic molecules brought in excited states under influence of an applied voltage and created in the regions indicated electron injection layer (EIL) and hole transport layer (HTL), respectively, indicated in Figure 1b. The polarons have an effective mobility based on their diffusion by hopping of the excitation from one molecule to the next. In the emitting layer (EL, see Figure 1b) both type of polarons will be present allowing them to recombine forming excitons, with 25% chance of a singlet exciton and 75% chance of a triplet. Only the singlet excitons can decay optically to the ground state, whereas the decay of the triplets to ground state is optically forbidden. The schematic energy level diagram corresponding to Figure 1a is depicted in Figure 1c. The hole-blocking role of TPBi can be explained with the HOMO energy difference between TPBi (HBL) and Alq3 (EL) being 0.5 eV whereas it is only 0.2 eV for the corresponding LUMO.

The layer structure is integrated with a horizontal cavity consisting of a second-order Bragg grating sandwiched between two first-order Bragg gratings, such that the photons are emitted downward due to the second-order grating. This configuration is sketched in Figure 1e, with a top-view photograph of the Bragg gratings in Figure 1d. The blue arrow starting from Figure 1b points to one of the various organic heterostructure units in Figure 1e that provide the optical gain in the cavity formed by the grating structure.

The optical gain is provided by the singlet excitons in the emitting layer. When a singlet decays radiatively, the molecule is left in the ground state and the exciton will disappear. There are several other decay channels for the singlets, that is, intersystem crossing (ISC), singlet-singlet annihilation (SSA), singlet-triplet annihilation (STA), and singlet-polaron annihilation (SPA). These decay processes will be explained briefly here; they are extensively discussed in [17]. ISC is a spin-flip induced intra-molecular process in which the singlet decays to the triplet on the same molecule, i.e., a loss of 1 singlet and at the same time a gain of 1 triplet. The other annihilation processes are of bi-molecular nature. In case of SSA, the interaction of two singlet excitons yields one ground state molecule plus one exciton with 25% chance of a singlet and 75% chance of a triplet, i.e., on average a net loss of 7/4 singlet and a net gain of 3/4 triplet [17]. In case of STA, the interaction of one singlet exciton and one triplet exciton leads to annihilation of the singlet exciton, which has decayed to the ground state, i.e., a net loss of 1 singlet. The interaction between a polaron

and a singlet exciton in case of SPA leads to annihilation of the singlet, both for hole-like and electron-like polarons.

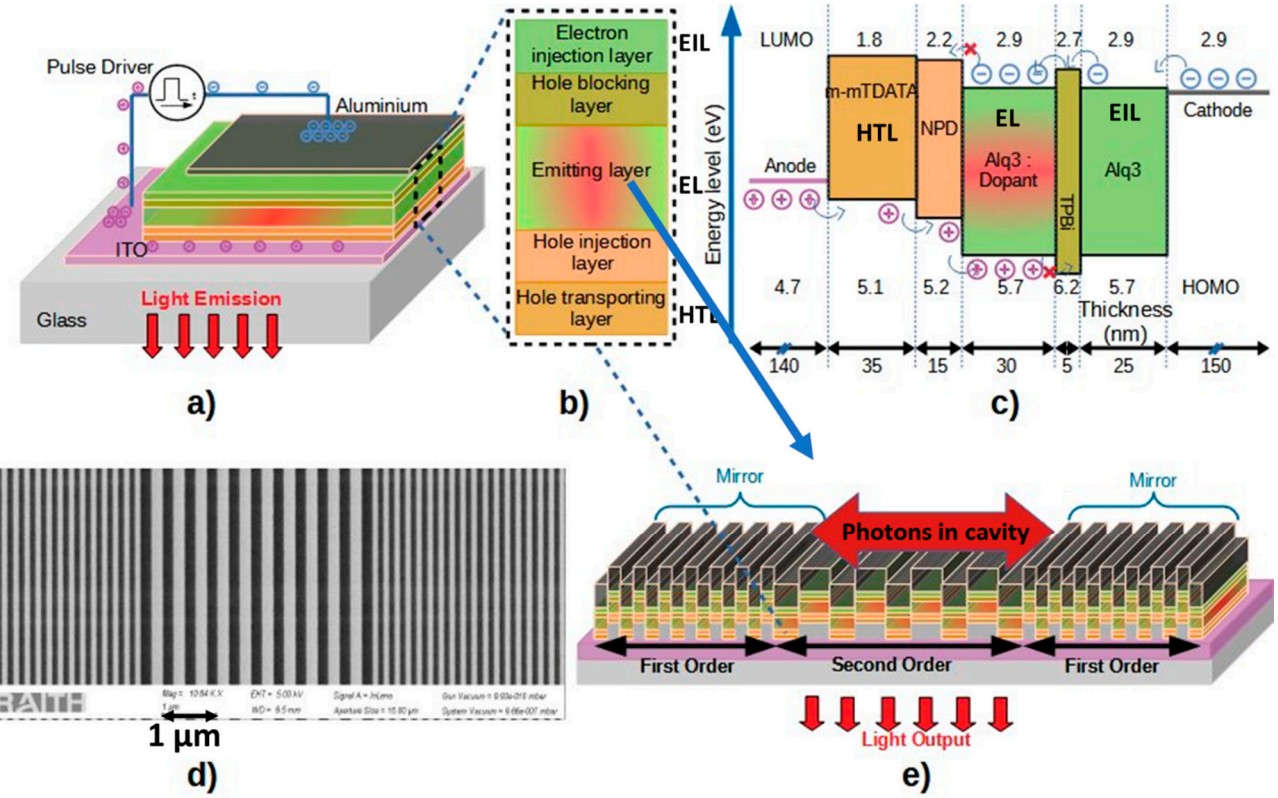

**Figure 1.** Schematics and direction of light emission of (**a**) the layer structure indicated in (**b**) with (**c**) the energy level diagram. In (**d**) a top view photograph is shown of the second-order grating sandwiched between the two first-order gratings and (**e**) presents a sketch of the OLD structure indicating the confined light between the mirrors formed by the first-order gratings.

In view of STA, proper accounting of the triplet population is crucial for the calculation of the optical gain (or: will be decisive for the available optical gain). Triplets are annihilated in the bi-molecular processes of triplet-triplet annihilation (TTA) and triplet-polaron annihilation (TPA). Like SSA, in TTA the interaction of two triplet excitons yields one ground state molecule plus one exciton with 25% chance of a singlet and 75% chance of a triplet. Hence, TTA leads, on average, to a net loss of 5/4 triplet and a net gain of 1/4 singlet. TPA leads to a loss of the triplet, just as SPA for the singlet does. In the rate equations that will be presented in Section 3, each of the above described processes corresponds to a term with corresponding rate coefficient.

## 3. Rate-Equation Model for the ODL

In the model we assume that the hole-type and electron-type polarons that participate in charge transfer across the organic semiconductor layers recombine in the emitting layer to form singlet and triplet excitons. We consider the situation where the emitting layer is composed of host molecules (the matrix), doped with a few percent of guest molecules (the dopant), and where the excitonic states are quickly transferred from the host molecules to the singlet and triplet excitons of the dopant molecules by Förster transfer and, to a lesser extent, by Dexter transfer, respectively. With a host-guest system like Alq3:DCM [18], the host (Alq3) singlets have their optical transition in the green part of the spectrum (~530 nm), whereas the dopant (DCM) singlets provide both spontaneous and stimulated emission in the red spectrum (~620 nm).

The model equations read

$$\frac{d}{dt}N_P = \frac{J(t)P_0}{ed} - \gamma N_P^2; \tag{1}$$

$$\frac{d}{dt}N_S = \frac{1}{4}\gamma N_P^2 + \frac{1}{4}\kappa_{TT}N_T^2 - (\kappa_{FRET}P_{0D} + \kappa_S + \kappa_{ISC})N_S - \left(\frac{7}{4}\kappa_{SS}N_S + \kappa_{SP}N_P + \kappa_{ST}N_T\right)N_S; \tag{2}$$

$$\frac{d}{dt}N_T = \frac{3}{4}\gamma N_P^2 + \kappa_{ISC}N_S + \frac{3}{4}\kappa_{SS}N_S^2 - (\kappa_{DEXT}P_{0D} + \kappa_T + \kappa_{TP}N_P)N_T - \frac{5}{4}\kappa_{TT}N_T^2; \tag{3}$$

$$\frac{d}{dt}N_{SD} = \kappa_{FRET}P_{0D}N_S + \frac{1}{4}\kappa_{TTD}N_{TD}^2 - (\kappa_{SD} + \kappa_{ISCD})N_{SD} -$$
$$\left(\frac{7}{4}\kappa_{SSD}N_{SD} + \kappa_{SPD}N_P + \kappa_{STD}N_{TD}\right)N_{SD} - \xi_E M(E, E, CAV)(N_{SD} - WN_{0D})P_{HO}; \tag{4}$$

$$\frac{d}{dt}N_{TD} = \kappa_{DEXT}P_{0D}N_T + \kappa_{ISCD}N_{SD} + \frac{3}{4}K_{SSD}N_{SD}^2 - \kappa_{TD}N_{TD} - \frac{5}{4}\kappa_{TTD}N_{TD}^2 - \kappa_{TPD}N_{TD}N_P; \tag{5}$$

$$\frac{d}{dt}P_{HO} = \beta_{sp}\kappa_{SD}N_{SD} + \{\Gamma\xi_E M(E, E, CAV)(N_{SD} - WN_{0D}) - \kappa_{CAV}\}P_{HO}; \tag{6}$$

$$N_0 = N_{HOST} - 2N_P - N_S - N_T; N_{0D} = N_{DOP} - N_{SD} - N_{TD}; \tag{7}$$

$$N_{DOP} = CN_{MOL}; N_{HOST} = (1 - C)N_{MOL};$$

$$P_0 = \frac{N_0}{N_{MOL}}; P_{0D} = \frac{N_{0D}}{N_{DOP}}. \tag{8}$$

These equations are valid in the emitting layer and the variables are: $N_P$ the polaron density, $N_S$ the density of singlet excitons, $N_T$ the density of triplet excitons, $N_0$ the density of ground-state molecules, all in the host; $N_{SD}$, $N_{TD}$ and $N_{0D}$ the respective dopant singlet, triplet and ground-state population densities. $P_{HO}$ is the photon density, $J(t)$ the current density, $N_{MOL}$ is the molecular density, $N_{DOP}$ and $N_{HOST}$ the respective densities of dopant and host molecules. $P_0$ and $P_{0D}$ are the respective probabilities that a host or dopant molecule is in the ground state. Finally, W represents the overlap between the dopant absorption spectrum $S_A(\lambda)$ and the emission spectrum $S_E(\lambda)$,

$$W \equiv \frac{\xi_A M(A, E, CAV)}{\xi_E M(E, E, CAV)} \equiv \frac{\xi_A \int d\lambda S_A(\lambda)S_E(\lambda)S_{CAV}(\lambda)}{\xi_E \int d\lambda S_E^2(\lambda)S_{CAV}(\lambda)}, \tag{9}$$

where $\xi_X$ are the coefficients for emission ($X = E$) and absorption ($X = A$) of the dopant and the normalization should be such that $\int d\lambda S_{CAV}(\lambda) = 1$ and $S_X(\lambda_X) = 1$, with $\lambda_X$ the wavelength for which $S_X$ is maximal ($X = E, A$). Note that W in (9) also depends on the cavity mode wavelength $\lambda_{CAV}$.

The derivation of (9) and the definition of *M* are given in Appendix A. W accounts for the fraction of dopant ground-state molecules that participate in the absorption of the emitted light. Note that $W = 1$ for identical spectra and $\xi_A = \xi_E$. The various parameters in (1) to (8) are listed in Table 1 together with their values. More about W will be discussed in Section 3.1.

Before we proceed with a brief discussion of the processes described by Equations (1)–(8), two remarks should be made. The first remark concerns the light emission by the host singlet excitons (green in case of Alq3). As we will see in Section 4, the build-up of $N_S$ remains relatively small, compared to $N_{SD}$. Moreover, no resonating structure is considered for the green light. Nevertheless, the host singlets will decay under spontaneous emission of green light. This photonic interaction is not considered in the rate equations.

As the second remark, note that the emission spectrum of the organic dopant emitter (DCM) is Stoke-shifted to the red by 160 nm from its absorption spectrum [19]. This implies that W will depend on the shift between the emission and absorption spectra as well as their respective widths. We estimate, using (A9) and (A10) (see Appendix A), that in the weak micro-cavity limit with $\kappa_{CAV} = 1.0 \times 10^{14} s^{-1}, Q \sim 18$, we estimate $W \approx 0.019$,

but as $\kappa_{CAV}$ decreases with increasing cavity quality factor and the threshold for lasing is approached, the emitted spectrum will narrow, implying smaller values for *W*. Therefore, *W* is a dynamic quantity, and this will be studied in more detail in Section 3.1.8.

**Table 1.** Model Parameters (Alq3:DCM).

| Symbol | Name | Value | Ref. |
|:---:|:---:|:---:|:---:|
| $S$ | OLED active area | $10^{-4}$ cm$^2$ | |
| $d$ | OLED active layer thickness | 30 nm | |
| $\gamma$ | Langevin recombination rate | $6.2 \times 10^{-12}$ cm$^3$ s$^{-1}$ to $2.0 \times 10^{-9}$ cm$^3$ s$^{-1}$ | [20,21] |
| $N_{MOL}$ | Molecular density | $2.1 \times 10^{21}$ cm$^{-3}$ | |
| $C$ | Dopant concentration | 2% | |
| $\kappa_{FRET}$ | Förster transfer rate | $1.15 \times 10^{10}$ s$^{-1}$ | [19,22,23] |
| $\kappa_{DEXT}$ | Dexter transfer rate | $1.0 \times 10^{10}$ s$^{-1}$ to $5.0 \times 10^{15}$ s$^{-1}$ | |
| $\kappa_S$ | Host singlet-exciton decay rate | $8.0 \times 10^7$ s$^{-1}$ | [24,25] |
| $\kappa_{SD}$ | Dopant singlet-exciton decay rate | $1.0 \times 10^9$ s$^{-1}$ | [24] |
| $\kappa_T$ | Host triplet decay rate | $6.5 \times 10^2$ s$^{-1}$ to $4.0 \times 10^4$ s$^{-1}$ | [24,26] |
| $\kappa_{TD}$ | Dopant triplet decay rate | $6.6 \times 10^2$ s$^{-1}$ | [26] |
| $\kappa_{ISC}$ | Host inter-system crossing rate | $2.2 \times 10^4$ s$^{-1}$ to $1.0 \times 10^7$ s$^{-1}$ | [17,27] |
| $\kappa_{ISCD}$ | Dopant inter-system crossing rate | $2.2 \times 10^4$ s$^{-1}$ to $1.0 \times 10^7$ s$^{-1}$ | [17,27] |
| $\kappa_{SS}$ | Host singlet-singlet annihilation (SSA) rate | $3.5 \times 10^{-12}$ cm$^3$ s$^{-1}$ | [24] |
| $\kappa_{SSD}$ | Dopant singlet-singlet annihilation (SSA) rate | $9.6 \times 10^{-13}$ cm$^3$ s$^{-1}$ | [24] |
| $\kappa_{SP}$ | Host singlet-polaron annihilation (SPA) rate | $3.0 \times 10^{-10}$ s$^{-1}$ | [24] |
| $\kappa_{SPD}$ | Dopant singlet-polaron annihilation (SPA) rate | $3.0 \times 10^{-10}$ cm$^3$ s$^{-1}$ | [24] |
| $\kappa_{TP}$ | Host triplet-polaron annihilation (TPA) rate | $2.8 \times 10^{-13}$ cm$^3$ s$^{-1}$ | [24] |
| $\kappa_{TPD}$ | Dopant triplet-polaron annihilation (TPA) rate | $5.6 \times 10^{-13}$ cm$^3$ s$^{-1}$ | [27] |
| $\kappa_{ST}$ | Host singlet-triplet annihilation (STA) rate | $1.9 \times 10^{-10}$ cm$^3$ s$^{-1}$ | [17,24] |
| $\kappa_{STD}$ | Dopant singlet-triplet annihilation (STA) rate | $1.9 \times 10^{-10}$ cm$^3$ s$^{-1}$ | [28,29] |
| $\kappa_{TT}$ | Host triplet-triplet annihilation (TTA) rate | $2.2 \times 10^{-12}$ cm$^3$ s$^{-1}$ | [24] |
| $\kappa_{TTD}$ | Dopant triplet-triplet annihilation (TTA) rate | $2.4 \times 10^{-15}$ cm$^3$ s$^{-1}$ | [27] |
| $\Gamma$ | Confinement factor | 0.29 | |
| $\xi_E$ | Dopant stimulated emission gain coefficient | $1.4 \times 10^{-5}$ cm$^3$ s$^{-1}$ | [17,20] |
| $\xi_A$ | Dopant absorption coefficient | $1.4 \times 10^{-5}$ cm$^3$ s$^{-1}$ | |
| $\kappa_{CAV}$ | Cavity photon decay rate | $1$–$300 \times 10^{12}$ s$^{-1}$ | |
| $\beta_{SP}$ | Spontaneous emission factor | <0.15 | |

## 3.1. Brief Discussion of Equations (1)–(8)

### 3.1.1. The Polaron Recombination

Polarons appear in two manifestations, positively charged hole-like polarons (density $N_P{}^+$) and negatively charged electron-like polarons (density $N_P{}^-$), where in view of assumed charge neutrality both populations are equal, $N_P{}^+ = N_P{}^-$. Moreover, each neutral polaron pair recombines to form one exciton together with one neutral molecule, which occurs at the Langevin-recombination rate $\gamma$ [17,30]. This recombination process drives the electrical current and leads to the sink term in (1). Since $\gamma$ is related to the polaron mobilities $\mu_h \wedge \mu_e$ as $\gamma = \frac{e}{\epsilon}(\mu_h + \mu_e)$, and since according to the Poole-Frenkel model the mobilities show an exponential dependence on the square root of the electric field *F*, we expect the value of $\gamma$ to increase substantially with increasing applied diode voltage. In ref. [20] the zero-field value $\gamma = 6.2 \times 10^{-12}$ cm$^3$ s$^{-1}$ is evaluated.

### 3.1.2. Host Singlet Excitons

The first term on the right-hand side (r.h.s.) of (2) is a source for the singlet excitons originating from the above-mentioned polaron recombination term. The factor 1/4 is due to the randomly injected spin statistics. The second term is a source term arising from triplet-triplet annihilation with generation rate $\kappa_{TT}$ [24]. All other terms are sink terms. The first sink term describes the Förster Resonance Energy Transfer (FRET) of singlet excitons

from host to dopant molecules with transfer rate $\kappa_{FRET}$. The probability $P_{0D}$ accounts for the potential depopulation of the dopant-ground state that would limit the energy transfer.

The second sink term describes the decay of the singlet exciton due to both radiative and non-radiative processes. The third sink term accounts for the inter-system crossing (ISC), a non-radiative mechanism, i.e., a spin-flip-induced intra-molecular energy transfer from singlet to triplet with a decay rate $\kappa_{ISC}$. The last sink terms in (2) describe the depopulation of the host singlet density with different annihilation terms: singlet-singlet annihilation (SSA) with decay rate $\kappa_{SS}$ [24], singlet-polaron annihilation (SPA) with decay rate $\kappa_{SP}$ [24], and singlet-triplet annihilation (STA) with decay rate $\kappa_{ST}$ [17,24].

### 3.1.3. Host Triplet Excitons

Rate Equation (3) describes the variation of host triplet excitons. The first three terms in the r.h.s. are sources. The first is a contribution arising from the polaron recombination. With a 3/4 factor resulting from the spin statistics, this source term, when added to the first singlet source term in (2), matches the first sink term for the polaron recombination in (1). The second term describes the increase of $N_T$ due to ISC in the same way as it decreases $N_S$ in (2). The third term corresponds to the decay of the triplet excitons with rate $\kappa_T$ [24,26]. The fourth and fifth terms correspond respectively to triplet-triplet annihilation (TTA) [24], and triplet-polaron annihilation (TPA) [24].

### 3.1.4. Dopant Singlet Excitons

The dynamics of the dopant-singlet density $N_{SD}$ is described by (4). The first term on the r.h.s. is the source because of the Förster energy transfer [19,22,23]. This term matches the corresponding sink term in (2). The second term is a relatively small and indirect source term due to the dopant triplet-triplet annihilation (TTA) leading to generation of singlets at rate $\kappa_{TTD}$ [27]. Except for the last term on the r.h.s., all other terms are the corresponding counterparts of terms in (2). In the first sink term, the dopant singlets decay radiatively at rate $\kappa_{SD}$. For the Alq3-DCM host-guest system we have taken the value $\kappa_{SD} = 1.0 \times 10^9$ s$^{-1}$ [24]. The last term describes the dopant singlet interaction with the photons due to stimulated emission with differential gain coefficient ξ.

### 3.1.5. Dopant Triplet Excitons

Rate Equation (5) describes the dopant triplet density $N_{TD}$ variations. The first term matches the corresponding Dexter transfer term in (3). The second term is the source resulting from the ISC matching the corresponding fourth term in (4). The third term represents the decay of the dopant triplet density at rate $\kappa_{TD}$ [26] by de-excitation, while other terms correspond to the absorption processes TTA ($\kappa_{TTD}$) and TPA ($\kappa_{TPD}$) [27].

### 3.1.6. Photons and Linewidth

Rate Equation (6) accounts for the dynamics of the photon density $P_{HO}$. The first term on the r.h.s. gives the spontaneous-emission contribution arising from the radiative recombination of the dopant singlets $N_{SD}$ at the rate $\kappa_{SD}$ where the spontaneous-emission factor $\beta_{sp}$ is the fraction of emitted photons within the lasing mode. The second term gives the net-amplification rate due to stimulated-emission

$$A_{STIM} \equiv \Gamma \xi_E M(E, E, CAV)(N_{SD} - WN_{0D}) - \kappa_{CAV}, \tag{10}$$

which will be large and negative so long the device operates below the lasing threshold but will climb up to a value close to zero if lasing is to be reached. In (10), $(N_{SD} - WN_{0D})$ is the effective inversion. $\Gamma$ is the confinement factor introduced to consider the fact that only the part of the photons inside the emitting layer is amplified. The last term on the r.h.s. accounts for the photon losses out of the cavity, with decay rate $\kappa_{CAV} = 1/\tau_{CAV}$, where

$\tau_{CAV}$ is the cavity photon lifetime. The dopant singlet density, for which the photon net loss, $WN_{0D} + \frac{\kappa_{CAV}}{\Gamma\xi_E M(E,E,CAV)}$, is precisely compensated, defines the threshold for lasing, i.e.,

$$N_{SD|thr} = WN_{0D} + \frac{\kappa_{CAV}}{\Gamma\xi_E M(E,E,CAV)}, \tag{11}$$

As we will see in Section 4, laser operation is characterized by the clamping of the dopant singlet density at a value very close to the value defined in (11) at the same time the net-amplification rate (11) is clamping near zero.

The frequency linewidth $\Delta\nu$ of the emitted light can be related to the effective photon cavity decay rate (see the last term in (6))

$$\kappa_{CAV,eff} = \kappa_{CAV} + \Gamma\xi_E M(E,E,CAV)(WN_{0D} - N_{SD}) \tag{12}$$

as [31]

$$\Delta\nu = \frac{\kappa_{CAV,eff}}{2\pi}, \tag{13}$$

valid as long the system is quasi cw and no linewidth enhancement due to amplitude-phase coupling occurs. The cavity width $\Delta_{CAV}$ that should be substituted in $M(E,E,CAV)$ and $W$ (see (A7), (A8), and (A11)), is related to the linewidth (13) as $\Delta_{CAV} = \frac{\lambda_E^2}{2\pi c}\Delta\nu$, with $c$ the vacuum light velocity.

### 3.1.7. Cavity Quality Factor

The outcoupling, diffraction, and absorption of the light in the cavity define a relationship between the cavity photon lifetime $\tau_{cav}$ and the corresponding quality factor $Q$ which reads:

$$Q = \omega_0\tau_{CAV}, \tag{14}$$

where $\omega_0$ is the resonance (angular) frequency of the cavity mode. The cavity photon decay rate $\kappa_{CAV}$ can be expressed in the quality factor $Q$ and the resonance wavelength in vacuum $\lambda_{CAV}$ as

$$\kappa_{CAV} = \frac{1}{\tau_{CAV}} = \frac{2\pi c}{n\lambda_0 Q}, \tag{15}$$

where $n$ is the refractive index. At 620 nm wavelength, a typical value for an OLED undergoing a parasitic weak microcavity is $Q \approx 6$, corresponding to a cavity decay rate of $\kappa_{CAV} \approx 3.0 \times 10^{14} \text{ s}^{-1}$. In a DFB-type laser cavity, a reasonable value for the quality factor $Q \approx 1800$ is achievable, and this corresponds to a cavity decay rate $\kappa_{CAV} \approx 1.0 \times 10^{12} \text{ s}^{-1}$.

### 3.1.8. (Re)Absorption Factor $W$

Despite the Stoke shift, the absorption spectrum $S_A(\lambda)$ and the emission spectrum $S_E(\lambda)$ of the emitted light by the dopant show some overlap, which induces a residual reabsorption of the photons emitted in the cavity by the dopant singlet excitons $N_{SD}$. With $W$ representing the spectral overlap, the reabsorption rate per unit photon density equals $\Gamma\xi_E M(E,E,CAV)WN_{0D}$. Note that the reabsorption of photons yields a source term for the dopant singlet population in (4). In the bad-cavity limit, the broad cavity spectrum $S_{CAV}$ and the absorption spectrum $S_A$ maximally overlap, hence re-absorption is maximal. When approaching the lasing threshold, the effective cavity spectrum narrows, and $W$ will assume its smallest value.

In Appendix A an expression for $W$ is derived in terms of integrals of intersecting spectra. For a model with Gaussian absorption and emission spectra, this expression can be written as (see (A8) and (A12))

$$W \equiv \frac{\xi_A M(A,E,CAV)}{\xi_E M(E,E,CAV)} = \frac{\xi_A C(A,E)\Delta_0(A,E)V(\lambda_{CAV} - \lambda_0(A,E);\Delta_0(A,E),\Delta_{CAV})}{\xi_E C(E,E)\Delta_0(E,E)V(\lambda_{CAV} - \lambda_0(E,E);\Delta_0(E,E),\Delta_{CAV})}, \tag{16}$$

where $V$ is the Voigt function [32], and definitions for $C(X, Y), \Delta_0(X, Y), \lambda_0(X, Y),$ and $\Delta_0(X, Y)$ are given in ((A13)–(A15)). Note that $W = 1$ in case of identical spectra ($\lambda_A = \lambda_E$, $\xi_A = \xi_E$ and $\Delta_A = \Delta_E$), irrespective of $\lambda_{CAV}$ and $\Delta_{CAV}$. Taking $\lambda_A = 460$ nm, $\Delta_A = 50$ nm, $\lambda_E = \lambda_{CAV} = 620$ nm and $\Delta_E = \Delta_{CAV} = 50$ nm, we find $W = 0.035$.

## 4. Simulations

### 4.1. Below Laser Threshold

We will first present results for an OLED without a special optical cavity, that is, for which $\kappa_{CAV} = 1 \times 10^{14}$ s$^{-1}$, corresponding to $Q \cong 18$. The current density of ~500 A/cm$^2$, switched on at time 0, is applied during 300 ns and the parameters are as in Table 1, except for some values mentioned in the caption of Figure 2. In Figure 2a, apart from the current density, time evolutions are seen for the ground-state probabilities $P_0$ for a host molecule, $P_{0D}$ for a dopant molecule, the spectral overlap $W$ and the linewidth $\Delta v$ of the light emitted by the dopant single excitons. $W$ and $\Delta v$ remain nearly constant at respective values $5 \times 10^{-2}$ and $1.6 \times 10^{13}$ s$^{-1}$. Due to the formation of excitons, the fraction of dopant molecules in the ground state falls to zero in ~100 ns, when the total number of dopant excitons approaches the total number of dopant molecules. This is demonstrated in Figure 2b, where it is seen that after ~100 ns the dopant triplet density $N_{TD}$ is already at the level of the total dopant density of $4.2 \times 10^{19}$ cm$^{-3}$. We recall that the light is emitted by the dopant singlet density $N_{SD}$ and the photon density $P_{HO}$ is seen to exhibit the same time development, apart from a proportionality constant. This is indicative for spontaneous emission. An important feature visible in Figure 2b is the sharp decrease of $N_{SD}$ after reaching its maximum ~7.5 ns after the current offset. The reason for this is the rapid increase of $N_{TD}$ and the associated singlet-triplet absorption STA leading to the dominant contribution to the singlet decay rate (see (4)) $\kappa_{STD} N_{TD} \sim 7.6 \times 10^9$ s$^{-1}$. Therefore, with the parameters as in Table 1, sufficient amount of gain to reach a laser threshold can only be expected in a small time interval below ~10 ns.

### 4.2. Validation of the Model for an OLED

To validate our model, we have confronted our simulation with an experimental analysis of an electrically pumped OLED without a special cavity. In this device, the organic hetero-structure itself defines a residual weak micro-cavity effect ($Q \sim 6$, $\kappa_{CAV} \sim 3.0 \times 10^{14}$ s$^{-1}$ and reabsorption fraction $W \sim 8\%$). A 20 ns, 45 V pulse excitation voltage is applied to the OLED and the electrical injection current is measured and recorded together with the emitted light intensity. This measured current is taken as the source term in the polaron rate Equation (1). The exciton and photon densities are then calculated from the set of Equations (1)–(8) with model parameters from Table 1, except for the fitted parameters given in Table 2. The results are plotted in Figure 3, where Figure 3d shows the measured and simulated photon densities in one plot for comparison.

The values for $\gamma$ and $\kappa_{DEXT}$ in Table 2 are the result of detailed fitting of the shape of the simulated photon response to the measured data. Variations in $\kappa_{DEXT}$ values mainly affect the leading edge and the maximum of the photon response. The black dashed curve in Figure 4d is the simulated photon density if the literature value $\gamma = 6.2 \times 10^{-12}$ cm$^3$ s$^{-1}$ instead of the fitted value in Table 2 is taken. The nearly two orders of magnitude larger value for $\gamma$ stems from the Poole-Frenkel effect for the mobility due to the internal electric field, induced by the high voltage of 45 V applied in the experiment (see Section 3.1.1). The validation of our model for a sub-threshold case is important, since it validates the gain behavior represented by the dopant singlet density $N_{SD}$. When the cavity quality factor is increased, the effective amplification rate by stimulated emission (see Equation (6)) will increase from large negative toward zero, without changing the underlying exciton dynamics, except $N_{SD}$ near and above the threshold. In fact, here the last term in Equation (4) will become the dominant loss, leading to clamping of $N_{SD}$ to the threshold value given in (10). Hence, we can use our validated model to predict laser operation behavior based on simulations where we increase the $Q$-factor of the OLED.

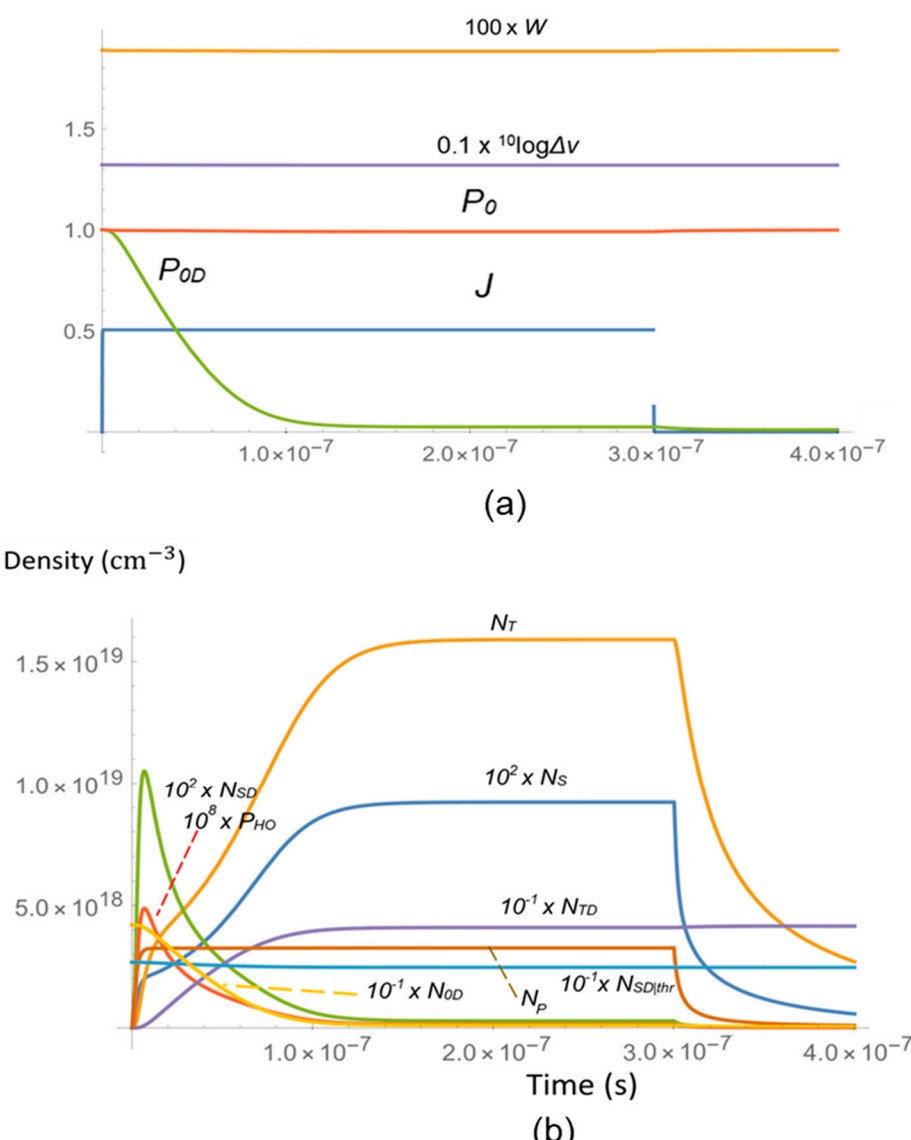

**Figure 2.** Time evolutions of various quantities for an OLED far below the threshold for lasing. The applied injection current has amplitude 0.5 kA/cm$^2$, and a duration of 300 ns. There is no special cavity arrangement assumed; $\kappa_{CAV} = 1 \times 10^{14}$ s$^{-1}$, corresponding to $Q \cong 18$, $\gamma = 1 \times 10^{-10}$ cm$^3$ s$^{-1}$, $\beta_{sp} = 0.05$, $\kappa_{DEXT} = 2 \times 10^8$ s$^{-1}$, $\kappa_T = 6.5 \times 10^2$ s$^{-1}$, $\kappa_{ISC(D)} = 2.2 \times 10^4$ s$^{-1}$. (**a**) Current density $J$ in kA/cm$^2$, ground-state probabilities $P_0$ and $P_{0D}$ respectively for host and dopant molecules and the linewidth $\Delta v$ of the emitted light. (**b**) Polaron density $N_P$, exciton densities for host and dopant molecules and the photon density $P_{HO}$. The light blue curve is $10^{-1} \times N_{SD}|_{thr}$ (see (11); clearly, the maximum of $N_{SD}$ (~ $10^{17}$ cm$^{-3}$) is far below the threshold value (~2.5 $\times 10^{19}$ cm$^{-3}$).

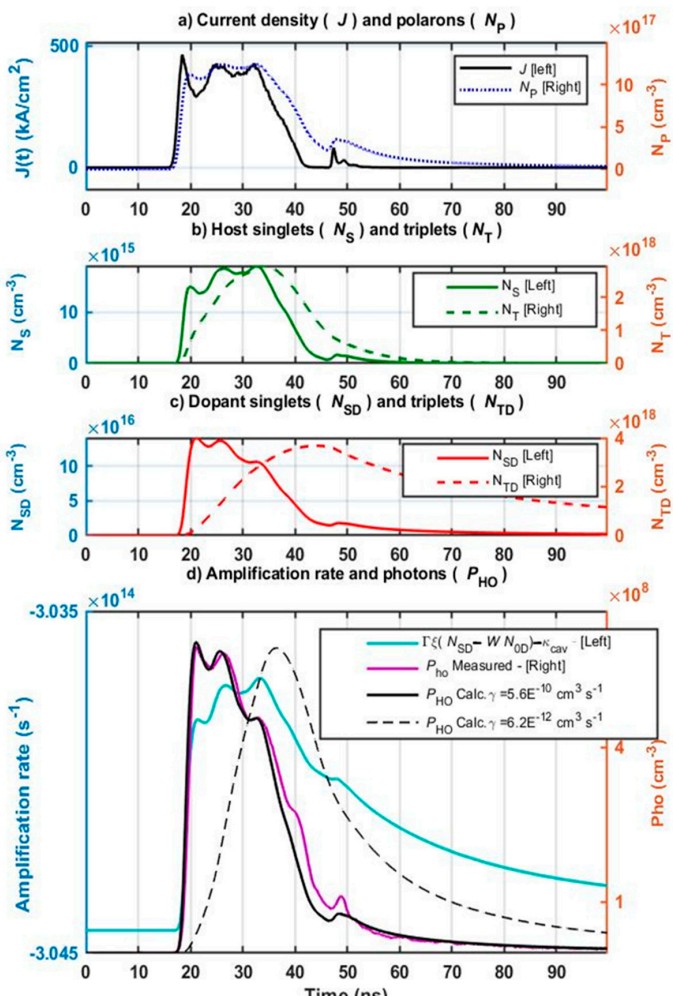

**Figure 3.** Comparison between measurements and simulation of the dynamical optical responses of an OLED under a peak current density of 0.462 kA/cm². In (**a**) the measured current and the simulated polaron density are depicted, in (**b**) the simulated host singlet and triplet densities and in (**c**) the dopant excitons. In (**d**) the measured (red curve) and simulated (black curve) photon densities (right scale) are shown together with the amplification rate (10) (blue curve; left scale). Parameters used in the simulation are given in Table 1, except for those given in Table 2. The dashed black curve in (**d**) is the simulated photon density for the zero-voltage value $\gamma = 6.2 \times 10^{-12}$ cm³ s⁻¹.

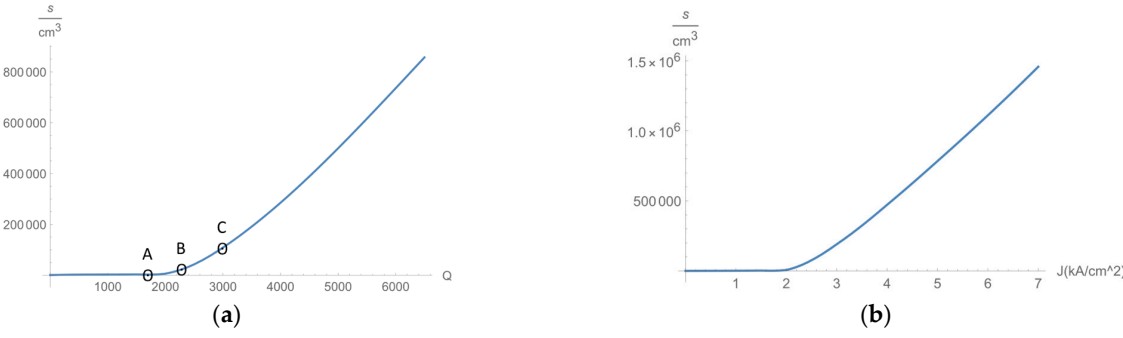

**Figure 4.** Simulated time-integrated photon density versus (**a**) the cavity quality factor $Q$ and (**b**) the applied electrical current density $J$. In (**a**) the pump current is 2.0 kA/cm² and the points A, B, C correspond to the cases of Figure 5a–c, respectively; in (**b**) the quality factor is 2000 ($\kappa_{CAV} \sim 9 \times 10^{11}$ s⁻¹). The duration of the pulse is 20 ns; the parameter values are for the validated case in Figure 3 and $\beta_{sp} = 3 \times 10^{-4}$.

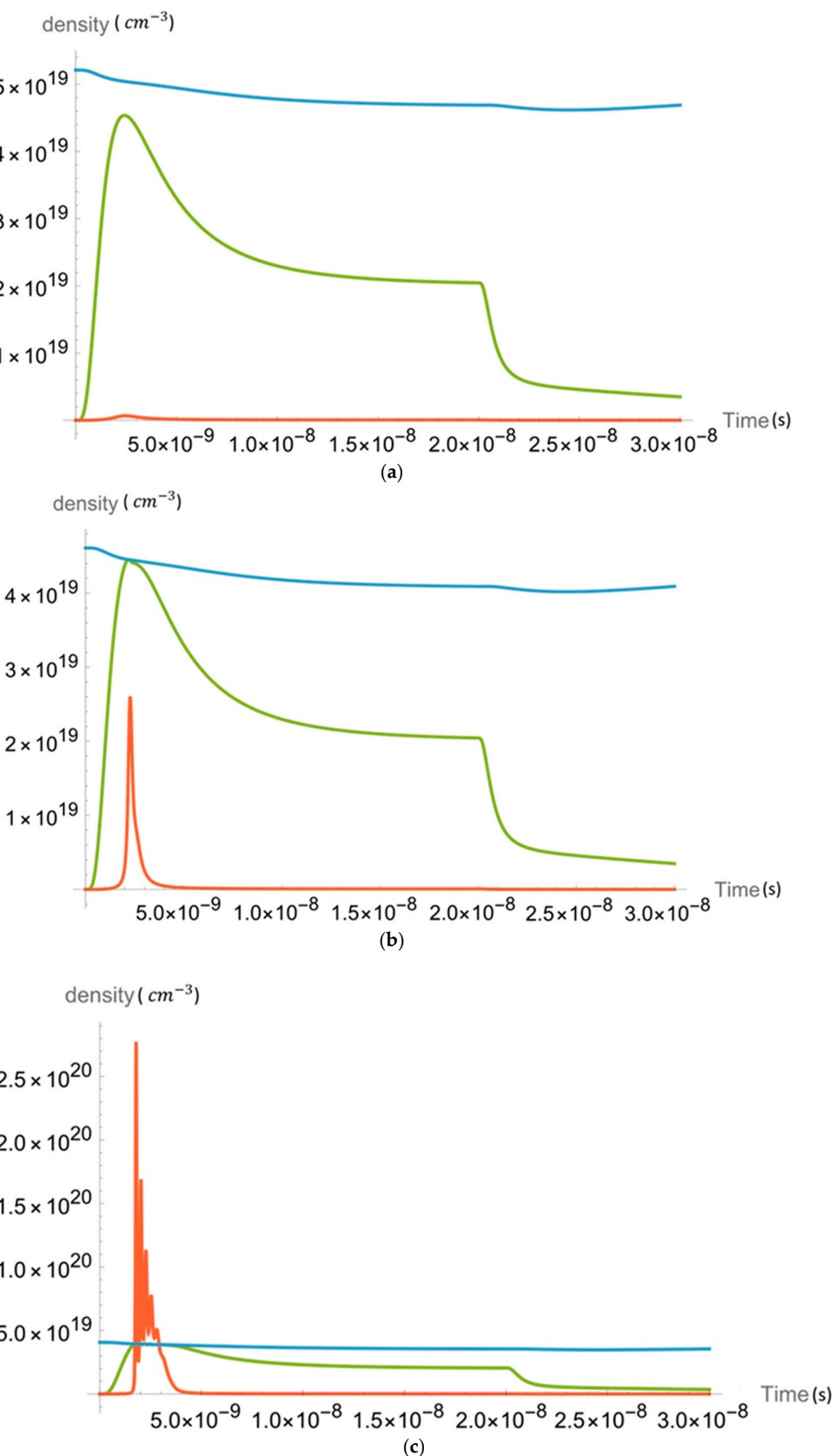

**Figure 5.** Time evolutions of the photon density $P_{HO}\left(\times 10^{6}\right)$ (red), singlet density $N_{SD}\left(\times 10^{2}\right)$ (green) and threshold density $N_{SD}|_{thr}\left(\times 10^{2}\right)$ (blue; see (11)) for (**a**) below threshold $Q = 1700$, (**b**) at threshold $Q = 2200$ and (**c**) above threshold $Q = 3000$. In each case the pump current density is 2 kA/cm$^2$ and applied from 0 to 20 ns while $\beta_{sp} = 3 \times 10^{-4}$.

**Table 2.** Model Parameters for Figure 3.

| Symbol | Name | Value |
|:---:|:---:|:---:|
| $S$ | OLED active area | $1.5 \times 10^{-4}$ cm$^2$ |
| $\gamma$ | Langevin recombination rate | $5.6 \times 10^{-10}$ cm$^3$ s$^{-1}$ |
| $\kappa_{DEXT}$ | Dexter transfer rate | $2.0 \times 10^{8}$ s$^{-1}$ |
| $\kappa_s$ | Host singlet-exciton decay rate | $8.3 \times 10^{7}$ s$^{-1}$ |
| $\kappa_T$ | Host triplet decay rate | $6.5 \times 10^{2}$ s$^{-1}$ |
| $\kappa_{ISC(D)}$ | Dopant inter-system crossing rate | $2.2 \times 10^{4}$ s$^{-1}$ |
| $\kappa_{SP}$ | Host SPA rate | $1.0 \times 10^{-11}$ s$^{-1}$ |
| $\kappa_{SPD}$ | Dopant SPA rate | $3.0 \times 10^{-10}$ cm$^3$ s$^{-1}$ |
| $\kappa_{TPD}$ | Dopant TPA rate | $9.0 \times 10^{-11}$ cm$^3$ s$^{-1}$ |
| $\kappa_{ST}$ | Host STA rate | $2.5 \times 10^{-10}$ cm$^3$ s$^{-1}$ |
| $\kappa_{STD}$ | Dopant STA rate | $3.7 \times 10^{-10}$ cm$^3$ s$^{-1}$ |
| $\kappa_{TTD}$ | Dopant TTA rate | $8.0 \times 10^{-12}$ cm$^3$ s$^{-1}$ |
| $\Gamma$ | Confinement factor | 0.29 |
| $\beta_{sp}$ | Spontaneous emission factor | $1.3 \times 10^{-3}$ |

*4.3. Laser Predictions*

So far, in the literature only one apparent though reserved and modest claim of observed lasing in an electrically pumped organic diode is given. It is reported in [33] and the OLED has the organic BsB-Cz as gain material in a configuration as sketched in Figure 1. Our model is validated for sub-threshold behavior but as we argued in the previous section, we can predict laser operation behavior by simulations with increasing *Q*-factor.

Figure 4a shows the simulated LQ-characteristic for the case of pump current $J = 2\,\mathrm{kA/cm^2}$. The pump pulse duration is 20 ns and the parameter values are for the validated case in Figure 3. A laser threshold is clearly seen at $Q_{TH} \sim 2200$ ($\kappa_{CAV} \sim 8.1 \times 10^{11}$ s$^{-1}$). In Figure 4b the integrated photon density versus the pump current is depicted for fixed quality factor $Q = 2000$ ($\kappa_{CAV} \sim 9.0 \times 10^{11}$ s$^{-1}$). This is the more usual LI-curve and shows the threshold at $J_{TH} \sim 2.2\,\mathrm{kA/cm^2}$. For the operation points labeled A, B, and C, the corresponding simulated photon density $P_{HO}(t)$, the singlet density $N_{SD}(t)$ and its threshold value $N_{SD\,thr}$ are plotted in the respective Figure 5a–c. In Figure 5a the maximum of the singlet density (green curve) remains below the threshold value for lasing (blue curve); in Figure 5b the singlet top just touches the threshold value at $t \sim 3$ ns producing a short laser pulse during ~1 ns. In Figure 5c, the system is above threshold, the singlet density is clamped to its threshold value from $t = 1.6$ ns up to $t = 3.9$ ns and during this time interval the system emits stimulated emission. The photon density after the onset of lasing shows a damped oscillation with frequency 3.8 GHz.

Such a relaxation oscillation is well known to occur in conventional III-V semiconductor lasers and more generally in class-B lasers [11,34]. For the case of Figure 5c, the evolutions of the reabsorption fraction $W(t)$ and the instantaneous linewidth $\Delta\nu$ (see (13)) of the emitted light are shown in Figure 6 together with the photon density oscillations. Note the 3 to 4 orders of magnitude linewidth reduction during the lasing phase.

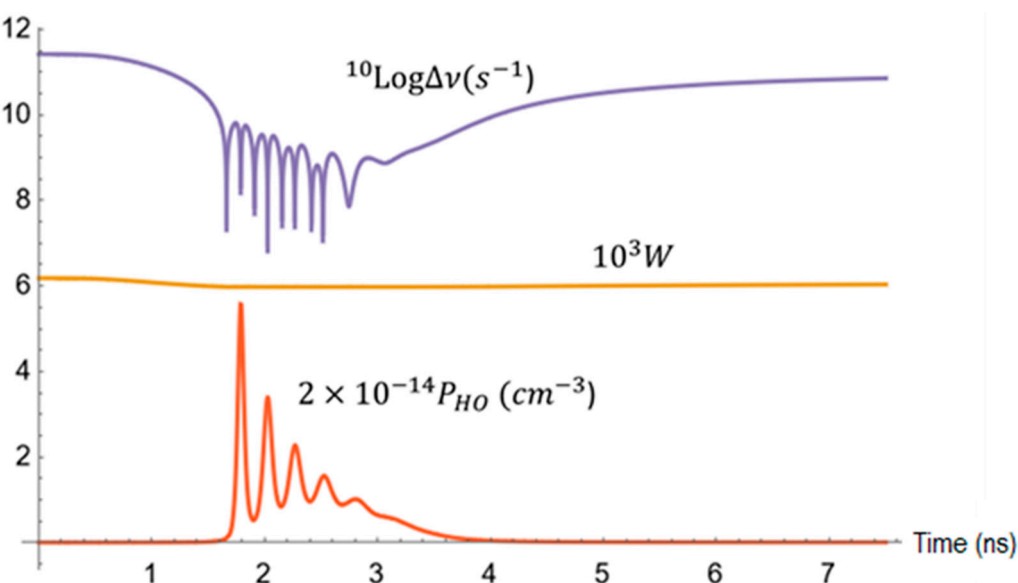

**Figure 6.** Time evolutions of the photon density (red), instantaneous linewidth (purple) and reabsorption factor (yellow) for the case of Figure 5c, i.e., $Q = 3000$ and $J = 2\ \text{kA/cm}^2$.

## 5. Discussion

With the formulation of the rate Equations (1)–(6) we have established the simplest possible model for an organic laser diode with a host-guest system that includes all known processes which underly the gain mechanism and the buildup of the photons in the cavity. The model applies to a host-guest system where the optical transitions of the host system can be disregarded and takes the reabsorption of the dopant into account in a self-consistent manner. The numerical simulations are for Alq3 as the host and a small volume fraction of 2% DCM as the dopant. Extension of the model to include light emitted from the host molecules is straightforward, and so is a different dopant fraction. The characteristic properties of the host and dopant molecules are reflected by the molecular parameter values, whereas the interaction of the gain with the optical field involves the optical-cavity parameters, such as the confinement and quality factor.

We have simulated the dynamics of the various molecular entities, i.e., the polarons and excitons responding to an electrical injection of $0.5\ \text{kA/cm}^2$ with a relatively long duration of 300 ns, which shows that a (quasi) steady state is reached after ~100 ns, which is characterized by a fully quenched gain. The reason is that the buildup of triplet excitons continues until all dopant molecules are used and no gain-providing singlets are present anymore. The bi-molecular interaction process of triplet-singlet absorption (TPA) starts to hamper the buildup of singlets already a few nanoseconds after the onset of the electrical injection. From this information it is concluded that if laser action with Alq3:DCM is to occur, it only will happen during a short time interval of a few nanoseconds.

The model has been validated by applying it to an experimental analysis of an electrically pumped OLED with weak residual micro-cavity effect defined by the organic heterostructure itself. The measured electrical current is taken as the source term in (1) and the emitted light intensity obtained with the model simulation is compared with the observed intensity. By fitting some of the parameters, good agreement was obtained (Figure 3d). Although in this case the emitted light is amplified spontaneous emission, we argue that based on the validation we can extrapolate to the case of laser emission. The argument is that most of the molecular dynamics remains unchanged; it is rather the cavity quality factor that will be different.

This leads to predictions for single-pulse laser operation during a few nanoseconds at most and accompanied by several orders of magnitude linewidth reduction and relaxation oscillations. For an applied current density amplitude of $2\ \text{kA/cm}^2$ the threshold can be

reached for $Q \sim 2200$, and for $Q \sim 2000$ the threshold current $J \sim 2.2\,\mathrm{kA/cm^2}$. These are all feasible values for practical systems.

Finally, we can speculate on the feasibility of CW laser operation. Inspection of Figure 5c suggests that if the singlet density $N_{SD}$ could be maintained at a larger value, it might be possible to extend the laser operation to a longer time interval. Apart from the singlet-exciton decay rates $\kappa_{S(D)}$, the parameters that influence the singlet decay most are the singlet-triplet absorption and to a lesser extent the singlet-polaron absorption rate. Indeed, assuming somewhat smaller values for $\kappa_{ST(D)}$ and $\kappa_{TP(D)}$ than in Table 2, we find CW laser operation with a linewidth of ~65 MHz in a simulation longer than 1000 ns electrical pumping for $J = 2.8\,\mathrm{kA/cm^2}$, $Q = 2000$ and $\kappa_{ST(D)} = 5 \times 10^{-11}\,\mathrm{cm^3\,s^{-1}}$, $\kappa_{SP(D)} = 1.0 \times 10^{-11}\,\mathrm{cm^3\,s^{-1}}$. For this case the laser threshold for CW operation is at ~ $0.8\,\mathrm{kA/cm^2}$. We have indications that these parameter values apply to the organic material BsB-Cz [16,33]. A systematic investigation of CW laser operation in dependance of molecular parameters will be published separately.

**Author Contributions:** Conceptualization and methodology, D.L., A.P.A.F. and M.C.; software, D.L. and A.P.A.F.; validation, A.P.A.F., A.O., A.C.C., N.L., and M.C.; formal analysis, D.L.; investigation, D.L. and A.P.A.F.; resources, A.P.A.F. and M.C.; data curation, A.O., A.C.C., N.L. and M.C.; writing—original draft preparation, D.L., A.P.A.F. and N.L.; writing—review and editing, D.L., A.P.A.F. and M.C.; supervision, A.P.A.F.; project administration, A.P.A.F.; funding acquisition, A.P.A.F. All authors have read and agreed to the published version of the manuscript.

**Funding:** This work was supported by the French Agence Nationale de la Recherche through the Program 453 Investissement d'Avenir under Grant ANR-11-IDEX- 0005-02, by the Labex SEAM: Science Engineering and 454 Advanced Materials. This study was also supported by the IdEx Université de Paris, ANR-18-IDEX-0001.

**Institutional Review Board Statement:** Not applicable.

**Informed Consent Statement:** Not applicable.

**Data Availability Statement:** The data presented in this study are openly available in https://filesender.renater.fr/?s=download&token=001c3249-cae6-4153-b3fe-6a7c072123ad.

**Acknowledgments:** The authors would like to thank J. Solard and D. Kocic for their technical support.

**Conflicts of Interest:** The authors declare no conflict of interest. The funders had no role in the design of the study; in the collection, analyses, or interpretation of data; in the writing of the manuscript, or in the decision to publish the results.

## Appendix A

To derive an expression for $W$, we decompose the total photon density in its cavity mode continuum. The photons in the active layer are distributed in wavelength according to the emission spectrum $S_E(\lambda)$ intersected with the normalized Lorentzian cavity profile $S_{CAV}(\lambda)$ with $\int d\lambda S_{CAV}(\lambda) = 1$. Hence

$$P_{HO}(\lambda) \equiv S_{CAV}(\lambda)S_E(\lambda)P_{HO}, \tag{A1}$$

$$P_{HO} = \int d\lambda P_{HO}(\lambda). \tag{A2}$$

The stimulated emission rate in the wavelength interval $d\lambda$ is

$$\dot{P}_{HO}(\lambda)|_{SE}d\lambda = \xi_E S_E(\lambda)N_{SD}P_{HO}(\lambda)d\lambda \tag{A3}$$

and the absorption is

$$\dot{P}_{HO}(\lambda)|_{ABS}d\lambda = \xi_A S_A(\lambda)N_{0D}P_{HO}(\lambda)d\lambda \tag{A4}$$

where $S_X(\lambda)$ is the emission respectively absorption spectrum for $X=E$, $A$, with

$$S_X(\lambda_X) = 1 \tag{A5}$$

and $\lambda_X$ is the wavelength for which the corresponding spectrum assumes its maximum value.

In principle, the respective linearized gain and absorption coefficients, $\xi_A$ and $\xi_E$, may be different. Subtracting (A4) from (A3), integrating over $\lambda$ and using (A2), the net stimulated emission can be written as

$$\dot{P}_{HO|stim} = \xi_E P_{HO} M(E,E,CAV) \left( N_{SD} - \frac{\xi_A M(A,E,CAV)}{\xi_E M(E,E,CAV)} N_{0D} \right), \tag{A6}$$

with

$$M(X,Y,CAV) \equiv \int d\lambda\, S_X(\lambda) S_Y(\lambda) S_{CAV}(\lambda), \ (X,Y = A,E). \tag{A7}$$

Hence, comparing (A6) with (6), we obtain

$$W \equiv \frac{\xi_A M(A,E,CAV)}{\xi_E M(E,E,CAV)} = \frac{\xi_A \int d\lambda\, S_A(\lambda) S_E(\lambda) S_{CAV}(\lambda)}{\xi_E \int d\lambda\, S_E^2(\lambda) S_{CAV}(\lambda)}. \tag{A8}$$

For numerical purposes, the following explicit forms for the spectra are introduced:

$$S_A(\lambda) = e^{-\frac{(\lambda - \lambda_A)^2}{2\Delta_A^2}} \ \text{(absorption spectrum approximated by a Gaussian);} \tag{A9}$$

$$S_E(\lambda) = e^{-\frac{(\lambda - \lambda_E)^2}{2\Delta_E^2}} \ \text{(emission spectrum approximated by a Gaussian).} \tag{A10}$$

Then, with the normalized Lorentzian cavity spectrum

$$S_{CAV}(\lambda) = \frac{\Delta_{CAV}}{\pi((\lambda - \lambda_{CAV})^2 + \Delta_{CAV}^2)}, \tag{A11}$$

the spectral overlap integral $M(X,Y,CAV)$ can be expressed in the Voigt function $V$ [32],

$$M(X,Y,CAV) = C(X,Y)\sqrt{2\pi}\Delta_0(X,Y)V(\lambda_{CAV} - \lambda_0(X,Y); \Delta_0(X,Y), \Delta_{CAV}), \tag{A12}$$

where $V$ is a standard built-in function and

$$C(X,Y) \equiv e^{-\frac{(\lambda||X - \lambda_Y)^2}{2(\Delta_X^2 + \Delta_Y^2)}}, \tag{A13}$$

$$\lambda_0(X,Y) \equiv \frac{\Delta_Y^2 \lambda_X + \Delta_X^2 \lambda_Y}{\Delta_Y^2 + \Delta_X^2}, \tag{A14}$$

$$\Delta_0(X,Y) \equiv \frac{\Delta_X \Delta_Y}{\sqrt{\Delta_X^2 + \Delta_Y^2}}. \tag{A15}$$

In case emission and absorption spectra are equal, i.e., $S_A(\lambda) = S_E(\lambda)$, $\xi_A = \xi_E$ and $\lambda_A = \lambda_E$ we find $W = 1$. With $\lambda_A = 460$ nm, $\Delta_A = 50$ nm, $\lambda_{CAV} = \lambda_E = 620$ nm, $\Delta_E = \Delta_{CAV} = 50$ nm, we calculate $W = 0.035$. For $\Delta_{CAV} \to 0$, i.e., very narrow cavity line ($Q \to \infty$), we find $W \to S_A(\lambda_E) = 0.006$. With $\Delta_{CAV} = \lambda_E^2 \kappa_{CAV}/(2\pi c)$ and $\kappa_{CAV} = 10^{14}$ s$^{-1}$, we find $\Delta_{CAV} = 20$ nm and $W = 0.019$.

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
