# Peer review of "Organic Diode Laser Dynamics: Rate-Equation Model, Reabsorption, Validation and Threshold Predictions"

_photonics, doi:10.3390/photonics8070279_

Round 1

Reviewer 1 Report

Lenstra et. al. in the manuscript ”Organic Diode Laser Dynamics: Rate-equation Model, Reabsorption, Validation and Threshold Predictions” demonstrated rate equation model to predict the lasing threshold in an electrically pumped OLD. The authors modeled six rate equations by considering all the probable exciton dynamics in the organic laser diode system. Most of the parameters have been borrowed from different references. In this manuscript, the authors tried to give a simplified lasing threshold prediction, but some aspects need further improvement.  My comments are given below;

Question 1: In the rate equation (2), the second term on the right-hand side (singlet generation factor from TTA) has been considered as 1/4*KTT*NT2. Why the singlet fraction generated out of TTA has to be 25% for this case is not clear. The maximum amount of singlet contribution from TTA in a fluorescent emitter could be determined from transient EL analysis at the pulse falling age(https://doi.org/10.1103/PhysRevB.85.045209, https://doi.org/10.1002/adom.201600678). Similarly what is the origin of 5/4 pre-factor in 5/4* KTT*NT2 depopulation term for triplets, is not clear.

              In the rate equation 2, the pre-factor in singlet depopulation rate via SSA has been considered as 7/4*KSS*Ns2. What is the origin of 7/4 fraction here? As the authors also described via the SSA process 3/4 fraction will contribute to triplet population. To me, this fractional contribution is not clear. Also, the proper reference for this is missing.

Question 2: The host and dopant both are fluorescent, then is it necessary to consider the ISC rate in modeling the rate equation? The monomolecular decay of singlets in both host and dopant should be much higher compared to the ISC rate.

Question 3: Among so many parameters, I feel it is necessary to define which parameters have been taken as variables and which are the fixed parameters for the model considered here?

Question 4: In Figure b, the authors pointed out the sharp decrease of dopant singlet density NSD due to the high rate of STA in the dopant. According to table 1, the STA rates in host and dopant are not very different. Then why STA is not causing host singlet to be decreased in a similar fashion? Unlike dopant, host singlet NS is seen to be gradually increasing.

Question 5: Since the dopant triplets can only be populated via Dexter energy transfer from the host triplet. It is necessary to know how KDex varies to control STA in the dopant.

Question 6: What is the reason for the observed damped oscillation in the polaronic current density in Figure 3?

Question 7: In figure 6, a strongly damped oscillation is observed in the lasing intensity for the injected current density of J=2kA/cm2. Wherefrom Figure 4b, it can be understood this is the threshold current density for which no significant damped oscillation has been shown in Figure 5b. Please, clarify the discrepancy.

Author Response

see file uploaded

Reviewer 2 Report

In the manuscript, D. Lenstra et al. reported a simple model based on six rate equations for an electrically pumped organic diode laser. On this basis, they applied this model into an organic host-guest system and predicted the threshold for short-pulse laser operation. The results are interesting and important. I can recommend the publication of this manuscript after the points noted below have been fully addressed in a suitable revision.

  1. In the introduction, when talking about organic lasers that cover the whole visible spectrum, some related literatures that modulating the lasing wavelength and broadening the lasing color range based on organic materials should be discussed extensively to catch broad interests. for reference: CCS Chem. 2021, 3, 624
  2. The x and y axes in Figure 2 and Figure 6 are suggested to be indicated to make them more informative to readers.

     3. Can this model be used to estimate the pump density of CW laser operation? If not, please state the differences between them.

     4. The font in all formulas should keep consistent.

     5. Finally, there are some errors in English usage in this paper that make it a bit unpleasant to read. For instance, all "guest-host" should replace with "host-guest" to keep consistency. The authors should carefully edit the manuscript to improve its quality.

Author Response

see file uploaded

Reviewer 3 Report

In this paper, a model for organic diode laser using rate equation is discussed and analyzed numerically. I think that the paper contains some new materials worth for publishing in Photonics. However, I made a few comments on the manuscripts.

  1. In Fig. 1, electron-like polarons pass through the potential barrier of TPBi, despite hole-like polarons cannot. I think that a brief explanation for this may help readers.
  2. In line 381, “Fig. 6.b” may be “Fig. 5.b”.
  3. What is the yellow line indicating in Fig. 5 ? The blue line in Fig. 6 is also unclear.
  4. The six rate equations may lead to complex and interesting dynamics depending on the device parameters. I am looking forward to the further reports.

Author Response

see file uploaded
